# Analysis of Circulating miRNA Expression Profiles in Type 2 Diabetes Patients with Diabetic Foot Complications

**DOI:** 10.3390/ijms25137078

**Published:** 2024-06-27

**Authors:** Giovanny Fuentevilla-Alvarez, María Elena Soto, Gustavo Jaziel Robles-Herrera, Gilberto Vargas-Alarcón, Reyna Sámano, Sergio Enrique Meza-Toledo, Claudia Huesca-Gómez, Ricardo Gamboa

**Affiliations:** 1Endocrinology Department, Instituto Nacional de Cardiología Ignacio Chávez, Juan Badiano No. 1. Col. Sección XVI, Mexico City 14080, Mexico; fuentevilla_alvarez@hotmail.com; 2Research Direction, Instituto Nacional de Cardiología Ignacio Chávez, Juan Badiano No. 1. Col. Sección XVI, Mexico City 14080, Mexico; mesoto50@hotmail.com (M.E.S.); gvargas63@yahoo.com (G.V.-A.); 3Cardiovascular Line in American British Cowdary (ABC) Medical Center, Sur 136 No. 116 Col. Las Américas, Mexico City 01120, Mexico; 4Phisiology Department, Instituto Nacional de Cardiología Ignacio Chávez, Juan Badiano No. 1. Col. Sección XVI, Mexico City 14080, Mexico; jaziel.gustavo@hotmail.com (G.J.R.-H.); c_huesca@yahoo.com (C.H.-G.); 5Coordinación de Nutrición y Bioprogramación, Instituto Nacional de Perinatología, Mexico City 11000, Mexico; ssmr0119@hotmail.com.mx; 6Biochemistry Department, Escuela Nacional de Ciencias Biológicas, Instituto Politécnico Nacional (IPN), Mexico City 11340, Mexico; semeza@hotmail.com

**Keywords:** microRNA, type 2 diabetes mellitus, diabetic foot

## Abstract

Type 2 diabetes mellitus (T2DM) is associated with various complications, including diabetic foot, which can lead to significant morbidity and mortality. Non-healing foot ulcers in diabetic patients are a major risk factor for infections and amputations. Despite conventional treatments, which have limited efficacy, there is a need for more effective therapies. MicroRNAs (miRs) are small non-coding RNAs that play a role in gene expression and have been implicated in diabetic wound healing. miR expression was analyzed through RT-qPCR in 41 diabetic foot Mexican patients and 50 controls. Diabetic foot patients showed significant increases in plasma levels of miR-17-5p (*p* = 0.001), miR-191-5p (*p* = 0.001), let-7e-5p (*p* = 0.001), and miR-33a-5p (*p* = 0.005) when compared to controls. Elevated levels of miR-17, miR-191, and miR-121 correlated with higher glucose levels in patients with diabetic foot ulcers (r = 0.30, *p* = 0.004; r = 0.25, *p* = 0.01; and r = 0.21, *p* = 0.05, respectively). Levels of miR-17 showed the highest diagnostic potential (AUC 0.903, *p* = 0.0001). These findings underscore the possible role of these miRs in developing diabetes complications. Our study suggests that high miR-17, miR-191, and miR-121 expression is strongly associated with higher glucose levels and the development of diabetic foot ulcers.

## 1. Introduction

Type 2 diabetes mellitus (T2DM) is a condition that predisposes individuals to other diseases, such as atherosclerosis, peripheral arterial disease, retinopathy, nephropathy, neuropathy, and impaired wound healing [1]. Diabetic foot is a complication of diabetes associated with significant morbidity [2]. A non-healing diabetic foot ulcer represents a major risk factor for infection or lower limb amputation [3]. The rates of lower extremity amputations secondary to diabetic foot were 8.3, 9.5, and 9.2 per 100,000 in the general population in Mexico in 2009, 2010, and 2011, respectively [4,5].

It is important to note that the 5-year prognostic mortality rate for patients with diabetic ulcers exceeds that of breast or prostate cancer [6]. Diabetic foot ulcers significantly contribute to both physical and psychological harm, making them a major public health issue. Conventional treatments have shown limited effectiveness in reducing the amputation rate, underscoring the need for more effective therapies. Several pathophysiological mechanisms are associated with impaired wound healing in diabetics, including dysfunctional ventricular remodeling independent of coronary artery disease, hypertension, and traditional cardiac risk factors [7,8]. Additionally, the deregulation of the extracellular matrix and the formation of reactive oxygen species play crucial roles in this impaired healing process [1]. Therefore, gaining a better understanding of the molecular mechanisms and biomolecules involved in the development of diabetic foot ulcers is essential for providing more effective therapeutic options for wound healing [9].

microRNAs (miRs) are short, small, non-coding RNAs, approximately 20 to 30 nucleotides in length. miRs have been considered as potential predictive, diagnostic, and even therapeutic biomarkers in many diseases, including diabetic wound healing [10,11,12]. Although miR alterations have been studied in diabetes [13,14,15], few studies exist in more severe stages of diabetic foot. It is well known that many factors may be regulating the severity of the disease; however, it is necessary to study the molecular mechanisms involved in its pathogenesis. Let-7e, miR-33a, and miR-144a are related to inflammation, immune response, and endothelial dysfunction, essential mechanisms in the pathogenesis of diabetic foot. These miRNAs can influence the expression of genes involved in chronic inflammation, oxidative stress, and altered immune response, thus contributing to the development and progression of diabetic foot complications [16,17]. In the case of miR-33a, it is well known to be associated with endothelial dysfunction and atherosclerosis in diabetic patients [18,19]. miR-144a has been related to the inflammatory response and angiogenesis, mechanisms involved in diabetic foot. It has been found that miR-144a regulates genes such as TNF-α and IL-6, suggesting an important role of this miR in the pathogenesis of the disease [20,21]. This study aimed to analyze the expression profiles of circulating miRNAs in the plasma of adult patients with well-established type 2 diabetes and the development of diabetic foot and compare them with the profiles in healthy individuals to identify specific miRNAs of diabetes. At the same time, we evaluate in silico metabolic pathways and biological processes related to the state of this disease.

## 2. Results

### 2.1. Characteristics of the Study Population

This study included 91 participants, of which 41 were diabetic patients, all with foot ulcers requiring debridement, and 50 healthy controls who had no diabetes, hypertension arterial (HTA), or subclinical atherosclerosis.

Table 1 shows the biochemical and anthropometric variables in the study groups. Significant differences were observed in some of these variables in patients with diabetic foot and the control group. In particular, significant differences were found in age (55.71 ± 13.50 vs. 47.8 ± 5.11, *p* = 0.0001), total cholesterol (228.54 ± 45.37 vs. 186.22 ± 37.57, *p* = 0.003), high-density lipoprotein cholesterol (HDL-C) (34.61 ± 2.95 vs. 46.48 ± 12.61, *p* = 0.001), low-density lipoprotein cholesterol (LDL-C) (132.71 ± 41.16 vs. 113.47 ± 32.25, *p* = 0.030), and blood glucose (134.78 ± 58.00 vs. 92.60 ± 8.86, *p* = 0.001).

### 2.2. miR Expression in Plasma

miR expression in plasma was significantly higher in the diabetic foot group compared with the control group in the case of let-7e-5p [control median: 18.11 (min. 0.30–max. 106.8) vs. diabetic foot median: 474.0 (min. 0.02–max. 2445.0); *p* = 0.001], miR-17-5p [control median: 25.42 (min. 2.26–max. 209.7) vs. diabetic foot median: 664.1 (min. 0.06–max. 2408.0); *p* = 0.001], miR-191-5p [control median: 24.20 (min. 0.36–max. 157.3) vs. diabetic foot median: 1835.0 (min. 0.05–max. 5407); *p* = 0.001], miR-33a-5p [control median: 16.93 (min. 1.32–max. 87.16) vs. diabetic foot median: 88.96 (min. 0.77–max. 658.4); *p* = 0.005], and miR144-3p [control median: 15.32 (min. 1.68–max. 126.9) vs. diabetic foot median: 47.65 (min. 0.09–max. 450.4); *p* = 0.32]; see Figure 1.

### 2.3. Correlation between miR Expression Levels and Biochemical and Anthropometric Values

A significant positive correlation between elevated glucose levels and miR-17 (r = 0.30, *p* = 0.004), miR-191 (r = 0.25, *p* = 0.01), and miR-121 (r = 0.21, *p* = 0.05) expression was observed. Furthermore, a positive association was found between higher levels of LDL and older age with the expression of miR-17 (r = 0.26, *p* = 0.01) for both parameters. In patients with diabetes mellitus, an inverse correlation was identified between low or normal levels of triglycerides and elevated levels of miR-144 (r = −0.33, *p* = 0.001).

### 2.4. Comorbidities and miR Expression

In this study, the levels of different microRNAs (miRs) were evaluated in patients with diabetes mellitus and healthy controls, distinguishing between subgroups with and without comorbidities such as obesity and systemic arterial hypertension (SAH). The expression values of miRs were normalized using logarithms and presented as medians (min.–max.).

We observed that levels of miR-17 and miR-191 were significantly elevated in patients with diabetes, both with and without obesity, compared to healthy controls (*p* < 0.001). Conversely, levels of let-7e and miR-33a were significantly reduced in patients with diabetes, both with and without obesity, compared to healthy controls (*p* < 0.001). Additionally, patients with obesity showed significantly lower levels of let-7e and miR-33a compared to patients without obesity (*p* < 0.001). No significant differences were observed in miR-144 levels between the studied groups.

Regarding SAH, it was found that patients with SAH and diabetes had higher levels of miR-17 and miR-191 compared to patients with diabetes without SAH (*p* = 0.003). Furthermore, patients with SAH and diabetes also showed higher levels of let-7e compared to patients with diabetes without SAH (*p* = 0.02). However, no significant differences were found in the levels of miR-33a and miR-144 between these groups (Table 2).

Table 3 presents the impact of obesity on the expression of various microRNAs (miRs) in both control and diabetic patient groups. In the control group, obesity did not significantly affect the expression of any of the analyzed miRs, as indicated by non-significant *p*-values (NS). This suggests that in healthy individuals, obesity alone does not alter the expression levels of these specific miRs.

However, the scenario is different for the diabetic patient group. Significant differences in miR expression were observed between obese and non-obese diabetic individuals. Specifically, the expression levels of miR-17, miR-191, let-7e, and miR-33a were markedly elevated in obese diabetic patients compared to their non-obese counterparts. The *p*-values associated with these differences were highly significant, with miR-17 (*p* = 0.000), miR-191 (*p* = 0.0001), let-7e (*p* = 0.0001), and miR-33a (*p* = 0.003).

We conducted a comparative analysis stratifying patients based on the presence or absence of systemic arterial hypertension (SAH) in individuals with diabetes and compared these groups to healthy individuals. Our findings revealed significant differences in the expression levels of four specific miRs across these groups. The detailed comparison is illustrated in Figure 2, which highlights the distinct miR expression profiles associated with SAH in diabetic patients versus those without SAH and healthy controls.

In Figure 2, the data are organized into three primary groups: diabetic patients with SAH, diabetic patients without SAH, and healthy individuals. Each miR’s expression level is represented, showing clear variations between the groups.

### 2.5. Association of Clinical Parameters and miR Expression with Diabetic Foot

Figure 3 shows the results of the ROC curve analysis for five specific miRs: let-7e, miR-17, miR-191, miR-33a, and miR-144. For each miRNA, the area under the curve (AUC) is shown as a measure of the miRNA′s discriminatory ability to distinguish between the groups of interest (in this case, patients with diabetes and controls). It also includes a 95% confidence interval (CI 95%) for the AUC and the *p*-value indicating whether the AUC is significantly different from 0.5 (which represents random performance). The results indicated that miR-17 had the highest AUC value of 0.903 (95% CI: 0.834–0.972) and a Youden index of 0.41. miR-144 showed the lowest diagnostic performance with an AUC of 0.560 (95% CI: 0.435–0.684) and a Youden index of 0.05. All miRs, except miR-144, showed significant *p*-values.

## 3. Discussion

Diabetic foot ulcers (DFU) are adverse complications since they imply pain and deterioration in quality of life [22].

In diabetes, there are vascular problems, neuropathy, and complex systemic damage when complicated by trauma or infection of wounds in the foot, which together interfere with the healing process. Many patients with diabetes mellitus (DM) are highly susceptible to diabetic hindlimb ischemia (DHI). In the literature, there is information about the miR-33 family, which is crucial in the regulation of lipid metabolism and inflammation [23,24,25,26], and it has therefore become an important target for the treatment of obesity and dyslipidemia [27], diabetes [28], and atherosclerosis [29]. In this study, we found that patients with diabetic foot had elevated levels of miR-33 versus control individuals, with a significant statistical difference of *p* = (0.001). Therefore, this finding is fundamental since the possibility that miR-33 participates in various biological processes related to energy, metabolism, and cell cycle regulation can confirm previous findings, where it has been mentioned that the deficient or deregulated state of miR-33 expression could significantly affect the regulatory mechanisms of glucose and lipids or cell proliferation.

In an experimental model in rats, it has been seen that miR-33-5p improves spinal cord injury (SCI) in rats since it inhibits the apoptosis of PC12 cells induced by LPS [30]. However, this has not been evaluated concerning healing and improvement in diabetic foot ulcers or lesions.

Serum microRNA-33 levels associated with elevated glucose levels in prediabetic, and diabetic patients is a finding that supports the possible role of miR-33 in monitoring the onset and progression of prediabetes or complicated states of diabetes, as occurs with patients with foot ulcers and deterioration or advancement in their healing, which will require evaluation through future clinical trials to clarify its mechanism and diagnostic feasibility.

MiR-17-5p is downregulated in DM and plays an essential role in vascular protection. The release of endothelial progenitor cell (EPC) exosomes (EPC-EX) contributes to vascular protection and ischemic tissue repair by transferring the miRs they contain.

In an experimental mouse model, it was evaluated whether the enrichment of EPC-EX with miR-17-5p (EPC-EXsmiR-17-5p) confers vascular and skeletal muscle protection in DHI in vitro and in vivo. In this model, it has been observed that the expression of miR-17-5p decreased markedly in the vessels and muscle tissue of the hind limbs and that infusing with EPC-EXsmiR-17-5p was more effective than EPC-EX for increased levels of miR-17-5p, as well as blood flow, microvessel density, and capillary angiogenesis, and reduces cell apoptosis [31].

Hyperglycemia is known to impair angiogenesis, and miR-17 has also been reported to participate in the proliferation, migration, and angiogenesis of a variety of vascular endothelial cells. Recent findings in mouse models of diabetes suggest that inhibition of miR-17 prevents high glucose (HG)-induced impairment of angiogenesis and improves cardiac function after MI in diabetic mice [32]. In this study about the expression of miR-17-5p, we found that subjects with diabetes had high glucose levels, which was noticeable in subjects with obesity and systemic arterial hypertension, and this is especially relevant given that it has been found to be associated with the disease.

Recently, it has been found that the upregulation of several miRNAs, such as miR-17-5p, miR-20b-5p, miR-29a-3p, and miR-126-3p, has improved the mutual phenomenon of gestational hypertension (GH) and serious pre-eclampsia (PE) [33]. Similarly, it has been proposed in other studies that epigenetic changes induced by pregnancy-related complications in placental tissue can cause the subsequent appearance of cardiovascular and cerebrovascular diseases in offspring [34].

Likewise, the participation of miR-17-5p and other molecular signatures derived from lung tissues has been highlighted that could elucidate the essential mechanisms of pulmonary arterial hypertension (PAH) as possible therapies associated with these molecules. This knowledge could help the development of new diagnostic tools and therapeutic strategies required to improve PAH.

For this reason, the study and understanding of miRs are particularly important since their behavior in a specific disease or, in this case, association with vasculopathy can lead us to understand better how these miRNAs function in cellular networks. This knowledge could help to define new molecular targets to propose therapies in diabetic vasculopathy [35].

The regulation and functions of miR-191 have allowed it to be proposed as a promising disease biomarker and therapeutic target.

miR-191-5p is predominantly expressed in platelets and endothelial cells [35,36] and modulates a wide range of cellular processes, such as proliferation, differentiation, migration, and apoptosis, and transcription factors associated with the cell cycle [37].

Measurement of circulating levels of miR-191-5p in people with conditions that alter the micro and macro vasculature is rare. However, decreased serum levels of miR-191-5p have been found in subjects with acute myocardial infarction [38,39] and in subjects with reinfarction [40]. Zampetaki et al. reported a reduction of miR-191-5p serum levels in patients with type 2 diabetes (T2DM) compared to controls [41,42]. Furthermore, miR-191-5p appears to delay wound healing in T2DM individuals with peripheral vascular disease by inhibiting both migration and angiogenesis. We found elevated levels of miR-191-5p, which is a finding that can be related to the experimental findings of Dangwai et al., who showed evidence that elevated miR-191 levels influenced angiogenesis and the migratory abilities of endothelial cells or diabetic dermal fibroblasts [43]

The dysfunction of miRNAs significantly contributes to the onset and progression of type 2 diabetes mellitus (T2DM), playing crucial roles in insulin secretion, glucose homeostasis, and adipocyte differentiation. Some circulating miRNAs are also implicated in the mechanisms underlying cardiovascular disease (CVD) [44,45]. Over the past decade, numerous miRNAs have been identified to play essential pathophysiological roles in T2DM. Let-7e has been associated with glucose metabolism, showing sensitivity to metformin treatment. Conversely, miR-144 has been linked to beta cell dysfunction and insulin signaling [46,47,48]. Despite advances in understanding the metabolic and vascular mechanisms in diabetes mellitus, the interactions between them are not yet fully defined [49].

On the other hand, miR-7e-5p has been suggested as a potential diagnostic marker for muscle atrophy, which could be relevant to diabetic vasculopathy [50]. Also, it has recently been found that in diabetic bone disease (DBD), which is a complication of diabetes mellitus (DM), there is impaired osteocyte function and delayed bone remodeling due to high blood glucose levels and the sustained release of inflammatory factors. And what has been found in studies of exosomes derived from bone marrow stromal cells (BMSCs) from non-diabetic subjects is that they greatly promote bone regeneration, while in exosomes derived from BMSCs from diabetics, they have the opposite effect of promoting MSC-Exos by regulating the miR-17/SMAD7 axis. These findings support the miR-17-5p/SMAD7 axis as a promising therapeutic target to treat DBD [51].

Our study revealed elevated levels of miR-7e and miR-144 in patients with diabetes, adding to the existing knowledge on the role of miRNAs in glucose metabolism, diabetes, and related complications. Contrary to previous findings, we did not observe an inverse association between high levels of miR-144 and normal or low levels of triglycerides in our cohort.

This finding is interesting because several miRNAs (including miR-144) have recently been reported to regulate different stages of lipid homeostasis, including biosynthesis, degradation, transport, and storage of low-density (LDL) and high-density cholesterol/fatty acids, as well as the formation of lipoproteins (HDL). [52]. There is also the hypothesis that the association of these miRNAs with insulin and the IGF growth factor system in receptors and binding proteins could represent a mechanism of regulation of the metabolic actions of IGF, which would be a topic of interest for investigations [53].

We mainly found in this study that obesity and high blood pressure are factors that influenced the results of miRs when compared with controls. In our controls, we had individuals with a high body mass index who tended to be overweight and obese. It was good that none of them had systemic arterial hypertension or dyslipidemia, which allowed us to see that when comparing the behavior of these risk factors between diabetic subjects and healthy subjects, it showed evidence of the apparent participation of comorbid risk factors in patients with DM and in the behavior of miR levels and their relationship to these associated diseases [54].

Studying the miRs involved in the development and progression of diabetic vasculopathy helps us understand the pathophysiology of vascular complications in diabetes mellitus and identify preventive therapeutic targets for treating diabetic vasculopathy. These miRs regulate the expression of target genes, significantly affecting VEGF signaling pathways (including *VEGFA*, *MAPK1*, *MAPK14*, and *PIK3CA*), insulin signaling, and metabolic processes (such as *IRS1*, *IRS2*, *IRS4*, *PIK3CA*, and *MTOR*). This regulation contributes to the pathogenesis and vascular complications of diabetes by modulating critical pathways involved in cell growth, proliferation, and survival.

The interaction network generated by MiRNet 2.0 reveals the complexity of these molecular interactions, showing how miRs and their target genes interact within these pathways. By mapping these interactions, the network highlights the central role of specific miRs in orchestrating the molecular events that lead to diabetic vasculopathy. Understanding this intricate web of interactions can aid in identifying key points for therapeutic intervention, making these miRs promising targets for the prevention and treatment of diabetic vasculopathy. Figure 4 provides a visual representation of these interactions, underscoring the potential impact of targeting these miRs in clinical applications.

## 4. Materials and Methods

### 4.1. Patient Population

A total of 91 unrelated Mexican subjects (41 patients with a diagnosis of essential type 2 diabetes mellitus with development of diabetic foot and 50 control subjects) were recruited at the Instituto Nacional de Cardiología Ignacio Chávez, of which 50% of the subjects were men and 50% women. A blood sample was taken from patients with diabetic foot ulcers requiring debridement of dead tissue. The inclusion criteria for both groups were as follows: being Mexican by birth with at least three previous generations of Mexican origin, being over 40 years old, and agreeing to participate by signing the informed consent. Furthermore, the control group should not have comorbidities. Controls were healthy asymptomatic individuals with no family history of T2DM, hypertension, or premature cardiovascular disease, recruited from the blood bank and through leaflets posted in Social Services centers. To ensure that patients in the control group did not have atheroma or subclinical atherosclerosis, their carotid intima-media thickness (cIMT) was assessed by ultrasound. All participants answered standardized and validated questionnaires to obtain information about their families, medical history, alcohol and tobacco use, and physical activity information. The clinical history of each subject was also revised. Research protocol 18-1075 was approved by the Research and Ethics Committees of the Institute. Suffering from a chronic degenerative disease, cancer, renal disease, familial hypertriglyceridemia, and autoimmune diseases were considered as exclusion criteria.

All participants signed written informed consent before inclusion in the study, and the study was approved by the Ethics Committee and complied with the Declaration of Helsinki [55].

### 4.2. Blood and Serum Samples

Blood samples were collected in sterile tubes containing EDTA from subjects who fasted for at least 12 h, during which they were only allowed to drink water. The serum was immediately separated by centrifugation for miR extraction and determination of the lipid profile [total cholesterol (TC), triglycerides (TG), high-density lipoprotein cholesterol (HDL-C), low-density lipoprotein cholesterol (LDL-C), and glucose.

### 4.3. Lipid Profile

Glucose, TC, and TG were assessed using enzymatic colorimetric methods (Roche-Syntex/Boehringer Mannheim, Mannheim, Germany). HDL-C was measured after the precipitation of low-density and very low-density lipoproteins with phosphotungstate/Mg^2+^ (Roche-Syntex), and LDL-C was estimated using the Friedewald equation [52], with modifications as detailed by De Long. All assays were conducted following an external quality control program (Lipid Standardization Program, Center for Disease Control in Atlanta, GA, USA). We adhered to the National Cholesterol Education Project (NCEP) is part of the National heart Lung and Blood Institute (NHLBI) Adult Treatment Panel (ATP III) guidelines and defined dyslipidemia based on the following levels: TC ≥ 200 mg/dL; LDL-C ≥ 130 mg/dL; HDL-C < 40 mg/dL for men and <50 mg/dL for women; and TG ≥ 150 mg/dL.

### 4.4. Definition

HDL-C concentration was considered abnormal if it was ≤0.9 mmol/L. TC ≥ 5.17 mmol/L was considered hypercholesterolemia, and levels of TG ≥ 1.69 mmol/L were considered abnormal [56]. Dyslipidemia was defined if an individual met at least one of the following criteria: TC ≥ 200 mg/dL, HDL-C ≤ 35 mg/dL, or TG ≥ 150 mg/dL [57].

### 4.5. miRs Extraction

Total RNA was purified from plasma samples using the miRNeasy Serum/Plasma Kit (Qiagen, Hilden, Germany) following the manufacturer′s instructions. As an internal control, a synthetic miRNA (cel-miR-39 from *C. elegans*) was added to each sample in equal amounts. Total RNA was stored at −80 °C.

### 4.6. Quantification of miRs by Real-Time PCR

The pulsed reverse transcription reaction was performed in serum to obtain cDNA of miR-17-5p, miR-191-5p, let-7e-5p, miR-144-3p, and miR-33a-5p with the specific primers for the mature forms, using the TaqMan miRNA RT Kit (TaqMan^®^ Advanced miRNA cDNA Synthesis Kit, Applied Biosystem, Foster City, CA, USA, Catalog Number A28007).

miR-17-5p (hsa-mir-17-5p), miR-191-5p (hsa-mir-191-5p), let-7e-5p (hsa-let-7e-5p), miR-144-3p (hsa-mir-144-3p), and miR-33a-5p (hsa-mir-33a-5p) were quantified using a commercial system kit (TaqMan gene expression assay) for microRNA, using the CFX96 real-time PCR system (BioRAD, Hercules, CA, USA). Conditions were 2 min at 50 °C and 10 min at 95 °C, followed by 40 cycles of 15 s at 95 °C and 1 min at 60 °C. Expression levels were measured in duplicate. and normalized to the endogenous gene cel-miR-39. The relative quantification was carried out using the following formula: 2^−ΔΔCt^ [58].

### 4.7. In Silico Analyses

To construct a network of microRNA (miRNA) interactions, we initially selected five relevant miRNAs (miR-17, miR-191, let-7e, miR-33a, and miR-144a) considering their known or suspected functions in the studied biological process or disease. We used the bioinformatics tool MiRNet 2.0 (developed and maintained at the University of Ottawa, Canada) to predict the target genes of these miRNAs by analyzing the sequence complementarity between the miRNAs and the mRNA [59]. With this information, we built the network where each miRNA was a node, and the interactions with its target genes were represented as edges. Subsequently, we analyzed the resulting network to identify interaction patterns, central nodes, and interaction modules, which allowed us to understand the gene regulation by the miRNAs in the specific biological context.

### 4.8. Statical Analyses

Data were analyzed using the SPSS v21 software (SPSS Inc., Chicago, IL, USA). The normality of each variable was determined with the Kolmogorov–Smirnov test. To represent the quantitative variables, the mean ± standard deviation (SD) was used, and the qualitative variables were reported as frequencies with percentages. Variables were compared using Student’s *t*-tests or Mann–Whitney U tests for 2 groups. We employed Pearson’s correlation analysis to assess the association between miR expression levels and clinical biochemical parameters. The diagnostic value of miR expression was assessed by calculating the area under the curve (AUC) in ROC models and Youden’s score index of each miR to define the optimal cutoff point.

### 4.9. Limitations

We did not conduct a follow-up evaluation in the patients to confirm whether the changes in glucose improved the levels of miRNAs, which does not allow us to confirm this hypothesis. However, this hypothesis can be raised in research through clinical trials and specific therapeutic measures to determine the importance of enhancing the levels of glucose or intervening directly in the control of miRNAs.

## 5. Conclusions

In this comprehensive analysis of miR expression in patients with diabetes mellitus, we observed a clear pattern of miR dysregulation associated with biochemical markers and clinical outcomes. Notably, miR expression levels were significantly elevated in patients with diabetic foot ulcers, indicating their potential role in the severity and progression of diabetes complications.

These findings underscore the importance of miRs as potential biomarkers for monitoring disease progression and complications in diabetic patients. The significant correlations with key metabolic parameters further suggest that miRs could be integral to understanding the molecular underpinnings of diabetes and its associated vascular and metabolic disturbances.

## Figures and Tables

**Figure 1 ijms-25-07078-f001:**
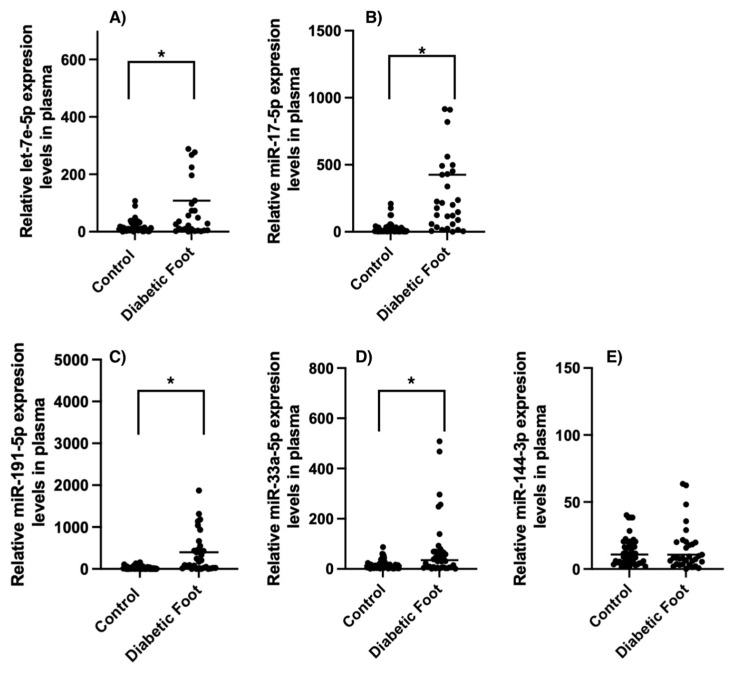
Comparison between control and diabetic foot patient expression levels in plasma of (**A**) let-7e-5p, (**B**) miR-17-5p, (**C**) miR-191-5p, (**D**) miR-33a-5p, and (**E**) miR144-3p expression. The data were normalized to RNU6B. The data are expressed as medians (min.–max.) (Mann–Whitney U test). * *p* > 0.05.

**Figure 2 ijms-25-07078-f002:**
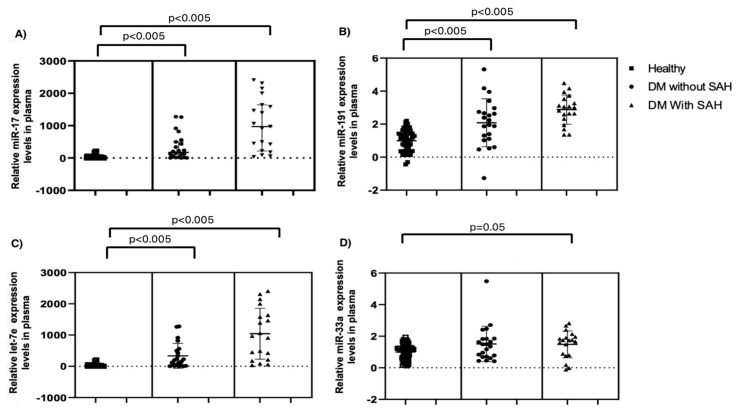
Comparison of miRs plasma expression in healthy controls, diabetic patients without SAH and with SAH: (**A**) miR-17-5p, (**B**) miR-191-5p, (**C**) let-7e-5p, and (**D**) miR-33a-5p expression. SAH: systemic arterial hypertension.

**Figure 3 ijms-25-07078-f003:**
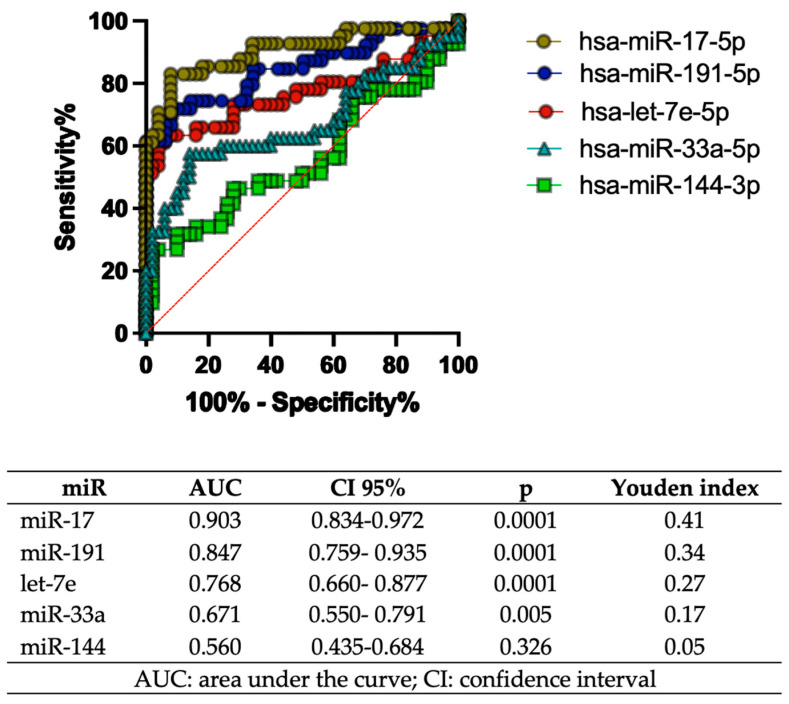
ROC curve analysis. An ROC model was performed to differentiate between the control and diabetic foot groups. The whole population was analyzed, 41 diabetic foot patients and 50 controls; the optimal cut-off point was defined through Youden’s score index to maximize the sum of sensitivity and specificity of miRs.

**Figure 4 ijms-25-07078-f004:**
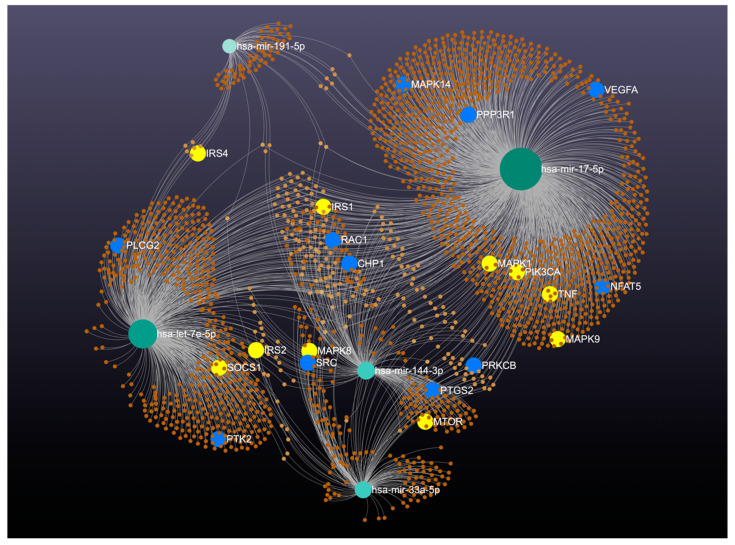
The interaction network generated by MiRNet 2.0 shows how five specific microRNAs: miR-17, miR-191, let-7e, miR-33a, and miR-144a, regulate key genes in the vascular endothelial growth factor (VEGF) signaling pathways and the development of diabetes. Each node represents a miRNA or a gene, with lines indicating regulatory interactions. Blue nodes mark genes associated with the VEGF signaling pathway, while yellow nodes highlight critical genes in the development of diabetes.

**Table 1 ijms-25-07078-t001:** Biochemical and anthropometric parameters of the study population.

Variables	Diabetic Foot (n = 41)	Control (n = 50)	*p*
Age	55.71 ± 13.50	47.8 ± 5.11	0.0001
Weight (kg)	76.48 ± 16.12	69.90 ± 11.59	0.028
BMI	28.92 ± 5.56	25.10 ± 2.91	0.001
Total Cholesterol (mg/dL)	228.54 ± 45.37	186.22± 37.57	0.001
HDL-C (mg/dL)	34.61 ± 2.95	46.48 ± 12.61	0.001
LDL-C (mg/dL)	132. 71 ± 41.16	113.47 ± 32.25	0.01
Triglycerides (mg/dL)	136.05 ± 18.48	137.28 ± 77.97	0.92
Glucose (mg/dL)	134.78 ± 58.00	92.60 ± 8.86	0.001
Diabetes % (n)	100%	0%	-
DMH	65%	0%	
HTA % (n)	57.5%	0%	-
SBP (mmHg)	124.46 ± 15.32	127.30 ± 20.22	0.476
DBP (mmHg)	73.78 ± 9.53	86.40 ± 20.40	0.001
HR (beats/min)	105.02 ± 30.21	77.02 ± 12.11	0.001
Smoking % (n)	24.8%	8.2%	-
Alcoholism % (n)	35%	69%	-

The data are expressed as mean ± SD (Student’s *t*-test) or as a percentage (chi-square test). BMI: body mass index; HDL-C: high-density lipoprotein cholesterol; LDL-C: low-density lipoprotein cholesterol; SBP: systolic blood pressure; DBP: diastolic blood pressure; HR: heart rate; HTA: hypertension arterial; DMH: diabetes mellitus history.

**Table 2 ijms-25-07078-t002:** miR levels in patients with diabetes mellitus and healthy controls in relation to comorbidities.

	Controls without Comorbidities	Patients with Diabetes Median (min.–max.) Q1, Q2, Q3				
		Without Obesity n = 12	With Obesity n = 25	p1	p2	p3
miR-17	0.82 (0.36–2.32) 0.54-0.82-1.50	2.34 (1.2–3.3) 2.1-2.3-3.1	2.69 (0.72–3.33) 1.7-2.6-3.1	0.000	0.000	0.000
miR-191	1.10 (−0.44–2.20) 0.37-1.10-1.45	2.6 (1.2–5.3) 1.6-2.6-3.8	2.6 (0.47–4.4) 1.6-2.6-3.0	0.0001	0.0001	0.000
let-7e	4.0 (−0.52–2.03) 0.71-1.0-1.48	2.3 (1.5–4.5) 1.3-2.3-3.2	1.9 (0.16–3.59) 0.99-1.9-2.6	0.0001	0.0001	0.001
miR33a	1.11 (0.12–1.94) 0.61-1.11-1.31	1.7 (0.44–2.8) 0.79-1.7-2.6	1.4 (0.02–5.4) 0.73-1.4-1.8	0.003	0.003	0.03
miR1-44	1.03 (0.23–2.10) 0.67-1.03-1.30	1.3 (2.0–2.6) 0.89-1.31-1.9	0.96 (0.13–2.5) 0.67-0.96-1.7	-	-	-
		**Without SAH n = 22**	**With SAH n = 19**			
miR-17	0.82 (0.36–2.32) 0.54-0.82-1.50	2.2 (1.2–3.1) 1.3-2.2-2.7	2.9 (1.5–3.3) 2.3-2.9-3.2	0.003	0.003	0.003
miR-191	1.10 (−0.44–2.20) 0.37-1.10-1.45	2.09 (1.2–5.3) 1.0-2.0-2.7	3.0 (1.3–4.4) 2.3-3.0-3.5	0.02	0.02	0.02
let-7e	4.0 (−0.52–2.03) 0.71-1.0-1.48	1.7 (1.5–4.5) 0.79-1.71-2.3	2.5 (0.27–3.59) 1.5-2.5-2.9	0.02	0.02	0.02
miR33a	1.11 (0.12–1.94) 0.61-1.11-1.31	1.4 (0.42–5.4) 0.70-1.4-1.8	1.7 (0.11–2.8) 0.79-1.74-1.96	-	-	-
miR1-44	1.03 (0.23–2.10) 0.67-1.03-1.30	0.91 (2.0–2.02) 0.67-0.91-1.2	1.4 (0.3–2.6) 0.78-1.4-2.0	0.03	-	0.0001
		**Without dyslipidemia n = 26**	**With dyslipidemia n = 15**			
miR-17	0.82 (0.36–2.32) 0.54-0.82-1.50	2.5 (1.2–3.3) 1.7-2.5-3.1	2.6 (0.72–3.3) 2.0-2.6-3.0	-	-	-
miR-191	1.10 (−0.44–2.20) 0.37-1.10-1.45	2.5 (1.2–4.1) 1.3-2.5-2.8	3.0 (0.47–5.32) 1.9-3.0-3.7	-	-	-
let-7e	4.0 (−0.52–2.03) 0.71-1.0-1.48	1.9 (1.5–3.3) 0.92-1.9-2.6	2.4 (0.16–4.5) 1.0-2.4-2.9	-	-	-
miR33a	1.11 (0.12–1.94) 0.61-1.11-1.31	1.2 (0.2–5.4) 0.73-1.2-1.8	1.7 (0.11–2.6) 1.4-1.7-1.8	-	-	-
miR1-44	1.03 (0.23–2.10) 0.67-1.03-1.30	2.1 (2.0–2.2) 0.77-1.1-1.5	0.98 (0.16–2.6) 0.53-0.98-2.0	-	-	-
		**Years of evolution <1** **–** **10**	**Evolution greater than 10 years**			
miR-17	0.82 (0.36–2.32) 0.54-0.82-1.50	2.3 (1.2–3.3) 1.8-2.3-2.9	2.6 (0.6–3.3) 2.1-2.6-3.1	-	-	-
miR-191	1.10 (−0.44–2.20) 0.37-1.10-1.45	2.5 (1.2–5.3) 1.3-2.5-3.0	2.8 (0.6–4.4) 1.8-2.8-3.6	-	-	-
let-7e	4.0 (−0.52–2.03) 0.71-1.0-1.48	1.9 (1.5–4.5) 0.82-1.9-2.6	2.3 (0.5–3.5) 1.1-2.3-2.9	-	-	-
miR33a	1.11 (0.12–1.94) 0.61-1.11-1.31	1.3 (0.11–2.8) 0.64-1.4-1.8	1.7 (0.18–5.4) 1.0-1.7-2.3	-	-	-
miR1-44	1.03 (0.23–2.10) 0.67-1.03-1.30	0.96 (2.0–2.6) 0.78-0.96-1.6	1.2 (0.13–2.5) 0.50-1.2-1.9	-	-	-

The expression values of miRs are normalized using logarithms. The values are presented as median (min.–max.) and the quartiles are 25, 50, and 75. SAH: systemic arterial hypertension. p1: controls vs. patients with diabetes without comorbidities; p2: controls vs. patients with diabetes and comorbidities; p3: diabetes without comorbidities vs. diabetes and comorbidities.

**Table 3 ijms-25-07078-t003:** Comparison of microRNA expression, stratified by obesity status.

	Control Median (min.–max.) Q1, Q2, Q3			Patients with Diabetes Median (min.–max.) Q1, Q2, Q3		
	Without Obesity N = 24	With Obesity N = 26	p1	Without Obesity N = 12	With Obesity N = 25	p2
miR-17	0.87 (0.36–2.32) 0.53-0.87-1.5	0.79 (0.36–2.259) 0.56-0.79-1.1	NS	2.34 (1.2–3.3) 2.1-2.3-3.1	2.69 (0.72–3.33) 1.7-2.6-3.1	0.000
miR-191	1.1 (0.13–2.2) 0.32-1.1-1.5	1.0 (0.44–2.1) 0.65-1.0-1.4	NS	2.6 (1.2–5.3) 1.6-2.6-3.8	2.6 (0.47–4.4) 1.6-2.6-3.0	0.0001
let-7e	1.1 (0.10–2.0) 0.78-1.1-1.4	0.92 (0.52–1.9) 0.63-0.92-1.2	NS	2.3 (1.5–4.5) 1.3-2.3-3.2	1.9 (0.16–3.59) 0.99-1.9-2.6	0.0001
miR33a	0.89 (0.15–1.78) 0.59-0.89-1.2	1.1 (0.12–1.9) 0.66-1.1-1.3	NS	1.7 (0.44–2.8) 0.79-1.7-2.6	1.4 (0.02–5.4) 0.73-1.4-1.8	0.003
miR1-44	1.0 (0.23–2.1) 0.86-1.0-1.2)	0.88 (0.28–1.6) 0.66-0.88-1.3	NS	1.3 (2.0–2.6) 0.89-1.31-1.9	0.96 (0.13–2.5) 0.67-0.96-1.7	NS

The expression values of miRs are normalized using logarithms. The values are presented as median (min.–max.) and the quartiles are 25, 50, and 75.

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
