# Peer review of "Analysis of Circulating miRNA Expression Profiles in Type 2 Diabetes Patients with Diabetic Foot Complications"

_ijms, 2024, doi:10.3390/ijms25137078_

Round 1
Reviewer 1 Report
Comments and Suggestions for Authors
The authors mention type 1 diabetic patients throughout this paper including the title, but the participants are type 2 diabetic patients. There are differences in foot complications resulting from type 1 and type 2 patients and so the paper needs rewording so that the focus is on type 2 only. Similarly, the authors use the terms diabetic foot "lesions" and "ulcers" interchangeably, e.g. most of the paper refers to ulcers, including the abstract, but the methodology asserts that the patients have been diagnosed with lesions. Whilst certainly similar, diabetic foot lesions and ulcers are different terms and the requirement for debridement is not specific to ulcers alone. The paper needs rewording to be more specific in this regard. Family histories were taken, but there is no mention in the study if confounding factors were taken into account in the participant selection. These details need to be included for the conclusions to be reliable. Can the authors be more specific as to the fasting period before serum sampling? The data and statistical comparisons in tables 2 and 3 are not explained well and do not make sense. How the authors conclude significant differences in the data presented in figure 2 is not adequately explained and not apparent from the data points in the graphs. The discussion is a series of "bullet point" sentences and short paragraphs that needs revision to appropriately put the results in context with the known field. Figure 4 appears thrown in without proper context (and referred to as figure 3 in the text).
Comments on the Quality of English LanguageThere are numerous syntax, grammar and punctuation mistakes throughout the manuscript which require the attention by an English language professional.
Author Response
Answer reviewer 1
We thank the reviewer for the time taken to review this research work. His valuable comments have undoubtedly increased the quality of this writing, and we have removed errors that the reviewer pointed out to us. Once again, the authors thank him for his suggestions.
- The authors mention type 1 diabetic patients throughout this paper including the title, but the participants are type 2 diabetic patients. There are differences in foot complications resulting from type 1 and type 2 patients and so the paper needs rewording so that the focus is on type 2 only.
Answer The authors apologize for the error. We focused on type 2 diabetes. We mistakenly included type 1 diabetes. As the reviewer rightly says, clinical manifestations vary between both types. We appreciate the observation.We have corrected it to type 2 diabetes. on the main page line 1
Similarly, the authors use the terms diabetic foot "lesions" and "ulcers" interchangeably, e.g. most of the paper refers to ulcers, including the abstract, but the methodology asserts that the patients have been diagnosed with lesions. Whilst certainly similar, diabetic foot lesions and ulcers are different terms and the requirement for debridement is not specific to ulcers alone.
Answet . The reviewer is correct, we have homogenized the terms for ulcers throughout the manuscript, specifically, in this work all the patients included in the study were patients with type 2 diabetes who presented ulcers that had to be debrided. The change was made on page 11 line 403
The paper needs rewording to be more specific in this regard. Family histories were taken, but there is no mention in the study if confounding factors were taken into account in the participant selection. These details need to be included for the conclusions to be reliable.
Answer we add this information in methodology thanks for the suggestion Furthermore, the control group should not have comorbidities. Controls were apparently healthy asymptomatic individuals. no family history of T2DM, hypertension or premature cardiovascular disease We made the change on page 12 line 406-408
Suffered from a chronic-degenerative disease, cancer, renal disease, familial hypertriglyceridemia and autoimmune diseases
Can the authors be more specific as to the fasting period before serum sampling?
Answer. Patients fasted for at least 12 h and it is allowed only to drink water. We made the change on page 12 line 423
The data and statistical comparisons in tables 2 and 3 are not explained well and do not make sense.
4A. We have corrected the table and added the columns of p separately, and we have also changed the description of the table
In this study, the levels of different microRNAs (miRs) were evaluated in patients with diabetes mellitus and healthy controls, distinguishing between subgroups with and without comorbidities such as obesity and systemic arterial hypertension (SAH). The expression values of miRs were normalized using logarithms and presented as medians (min-max).
We observed that levels of miR-17 and miR-191 were significantly elevated in patients with diabetes, both with and without obesity, compared to healthy controls (p<0.001). Conversely, levels of Let-7e and miR-33a were significantly reduced in patients with diabetes, both with and without obesity, compared to healthy controls (p<0.001). Additionally, patients with obesity showed significantly lower levels of Let-7e and miR-33a compared to patients without obesity (p<0.001). No significant differences were observed in miR-144 levels between the studied groups.
Regarding SAH, it was found that patients with SAH and diabetes had higher levels of miR-17 and miR-191 compared to patients with diabetes without SAH (p=0.003). Furthermore, patients with SAH and diabetes also showed higher levels of Let-7e compared to patients with diabetes without SAH (p=0.02). However, no significant differences were found in the levels of miR-33a and miR-144 between these groups. We made the change on page 4 lines 149-165
We change explanation of table 3 we add:
Table 3 presents the impact of obesity on the expression of various microRNAs (miRs) in both control and diabetic patient groups. In the control group, obesity did not significantly affect the expression of any of the analyzed miRs, as indicated by non-significant p-values (NS). This suggests that in healthy individuals, obesity alone does not alter the expression levels of these specific miRs.
However, the scenario is different for the diabetic patient group. Significant differences in miR expression were observed between obese and non-obese diabetic individuals. Specifically, the expression levels of miR-17, miR-191, let-7e, and miR-33a were markedly elevated in obese diabetic patients compared to their non-obese counterparts. The p-values associated with these differences were highly significant, with miR-17 (p=0.000), miR-191 (p=0.0001), let-7e (p=0.0001), and miR-33a (p=0.003). We made the change on page 6 lines 172-184
How the authors conclude significant differences in the data presented in figure 2 is not adequately explained and not apparent from the data points in the graphs.
Answer We conducted a comparative analysis stratifying patients based on the presence or absence of systemic arterial hypertension (SAH) in individuals with diabetes and compared these groups to healthy patients. Our findings revealed significant differences in the expression levels of four specific miRs across these groups. The detailed comparison is illustrated in Figure 2, which highlights the distinct miR expression profiles associated with SAH in diabetic patients versus those without SAH and healthy controls.
In Figure 2, the data is organized into three primary groups: diabetic patients with SAH, diabetic patients without SAH, and healthy individuals. Each miR’s expression level is represented, showing clear variations between the groups. We made the change on page 7 lines 205-219
The discussion is a series of "bullet point" sentences and short paragraphs that needs revision to appropriately put the results in context with the known field.
Answer . thanks for the suggestion we have corrected it in all discussion. They were grammar changes
Figure 4 appears thrown in without proper context (and referred to as figure 3 in the text).
6 Answer . We have corrected it in the text
We add the next information
Studying the miRs involved in the development and progression of diabetic vasculopathy helps us understand the pathophysiology of vascular complications in diabetes mellitus and identify preventive therapeutic targets for treating diabetic vasculopathy. These miRs regulate the expression of target genes, significantly affecting VEGF signaling pathways (including VEGFA, MAPK1, MAPK14, and PIK3CA), insulin signaling, and metabolic processes (such as IRS1, IRS2, IRS4, PIK3CA, and MTOR). This regulation contributes to the pathogenesis and vascular complications of diabetes by modulating critical pathways involved in cell growth, proliferation, and survival.
The interaction network generated by MiRNet 2.0 reveals the complexity of these molecular interactions, showing how miRs and their target genes interact within these pathways. By mapping these interactions, the network highlights the central role of specific miRs in orchestrating the molecular events that lead to diabetic vasculopathy. Understanding this intricate web of interactions can aid in identifying key points for therapeutic intervention, making these miRs promising targets for the prevention and treatment of diabetic vasculopathy. Figure 4 provides a visual representation of these interactions, underscoring the potential impact of targeting these miRs in clinical applications. This change was made on page 11 of line 372-390.

Reviewer 2 Report
Comments and Suggestions for Authors
MicroRNAs (miRs) are small non-coding RNAs that play a role in gene expression and have been implicated in diabetic wound healing. The authors suggest elevated expression of miR-17, miR-191 and miR-121 are strongly associated with high glucose levels and presence of the diabetic foot ulcer. This manuscript is interesting, but there are several concerns that should be addressed in the manuscript.
1. The authors showed that miR-33 level is elevated in the patients with diabetic foot. They should discuss the relationship between miR-33 and glucose levels and/or diabetic wound healing.
2. The authors describe that the inhibition of miR-17 prevents diabetic-induced impairment of angiogenesis. Does the improvement of glucose level reduce the level of miR-17?
3. I am interesting with the relationship between miRNA expression and SAH. The authors should discuss about that.
4. Is there the association between miR-144 level and lipid metabolism?
Comments on the Quality of English LanguageEnglish is generally well written.
Author Response
Answer reviewer 1
We thank the reviewer for the time taken to review this research work. His valuable comments have undoubtedly increased the quality of this writing, and we have removed errors that the reviewer pointed out to us. Once again, the authors thank him for his suggestions.
- The authors mention type 1 diabetic patients throughout this paper including the title, but the participants are type 2 diabetic patients. There are differences in foot complications resulting from type 1 and type 2 patients and so the paper needs rewording so that the focus is on type 2 only.
Answer The authors apologize for the error. We focused on type 2 diabetes. We mistakenly included type 1 diabetes. As the reviewer rightly says, clinical manifestations vary between both types. We appreciate the observation.We have corrected it to type 2 diabetes. on the main page line 1
Similarly, the authors use the terms diabetic foot "lesions" and "ulcers" interchangeably, e.g. most of the paper refers to ulcers, including the abstract, but the methodology asserts that the patients have been diagnosed with lesions. Whilst certainly similar, diabetic foot lesions and ulcers are different terms and the requirement for debridement is not specific to ulcers alone.
Answet . The reviewer is correct, we have homogenized the terms for ulcers throughout the manuscript, specifically, in this work all the patients included in the study were patients with type 2 diabetes who presented ulcers that had to be debrided. The change was made on page 11 line 403
The paper needs rewording to be more specific in this regard. Family histories were taken, but there is no mention in the study if confounding factors were taken into account in the participant selection. These details need to be included for the conclusions to be reliable.
Answer we add this information in methodology thanks for the suggestion Furthermore, the control group should not have comorbidities. Controls were apparently healthy asymptomatic individuals. no family history of T2DM, hypertension or premature cardiovascular disease We made the change on page 12 line 406-408
Suffered from a chronic-degenerative disease, cancer, renal disease, familial hypertriglyceridemia and autoimmune diseases
Can the authors be more specific as to the fasting period before serum sampling?
Answer. Patients fasted for at least 12 h and it is allowed only to drink water. We made the change on page 12 line 423
The data and statistical comparisons in tables 2 and 3 are not explained well and do not make sense.
4A. We have corrected the table and added the columns of p separately, and we have also changed the description of the table
In this study, the levels of different microRNAs (miRs) were evaluated in patients with diabetes mellitus and healthy controls, distinguishing between subgroups with and without comorbidities such as obesity and systemic arterial hypertension (SAH). The expression values of miRs were normalized using logarithms and presented as medians (min-max).
We observed that levels of miR-17 and miR-191 were significantly elevated in patients with diabetes, both with and without obesity, compared to healthy controls (p<0.001). Conversely, levels of Let-7e and miR-33a were significantly reduced in patients with diabetes, both with and without obesity, compared to healthy controls (p<0.001). Additionally, patients with obesity showed significantly lower levels of Let-7e and miR-33a compared to patients without obesity (p<0.001). No significant differences were observed in miR-144 levels between the studied groups.
Regarding SAH, it was found that patients with SAH and diabetes had higher levels of miR-17 and miR-191 compared to patients with diabetes without SAH (p=0.003). Furthermore, patients with SAH and diabetes also showed higher levels of Let-7e compared to patients with diabetes without SAH (p=0.02). However, no significant differences were found in the levels of miR-33a and miR-144 between these groups. We made the change on page 4 lines 149-165
We change explanation of table 3 we add:
Table 3 presents the impact of obesity on the expression of various microRNAs (miRs) in both control and diabetic patient groups. In the control group, obesity did not significantly affect the expression of any of the analyzed miRs, as indicated by non-significant p-values (NS). This suggests that in healthy individuals, obesity alone does not alter the expression levels of these specific miRs.
However, the scenario is different for the diabetic patient group. Significant differences in miR expression were observed between obese and non-obese diabetic individuals. Specifically, the expression levels of miR-17, miR-191, let-7e, and miR-33a were markedly elevated in obese diabetic patients compared to their non-obese counterparts. The p-values associated with these differences were highly significant, with miR-17 (p=0.000), miR-191 (p=0.0001), let-7e (p=0.0001), and miR-33a (p=0.003). We made the change on page 6 lines 172-184
How the authors conclude significant differences in the data presented in figure 2 is not adequately explained and not apparent from the data points in the graphs.
Answer We conducted a comparative analysis stratifying patients based on the presence or absence of systemic arterial hypertension (SAH) in individuals with diabetes and compared these groups to healthy patients. Our findings revealed significant differences in the expression levels of four specific miRs across these groups. The detailed comparison is illustrated in Figure 2, which highlights the distinct miR expression profiles associated with SAH in diabetic patients versus those without SAH and healthy controls.
In Figure 2, the data is organized into three primary groups: diabetic patients with SAH, diabetic patients without SAH, and healthy individuals. Each miR’s expression level is represented, showing clear variations between the groups. We made the change on page 7 lines 205-219
The discussion is a series of "bullet point" sentences and short paragraphs that needs revision to appropriately put the results in context with the known field.
Answer . thanks for the suggestion we have corrected it in all discussion. They were grammar changes
Figure 4 appears thrown in without proper context (and referred to as figure 3 in the text).
6 Answer . We have corrected it in the text
We add the next information
Studying the miRs involved in the development and progression of diabetic vasculopathy helps us understand the pathophysiology of vascular complications in diabetes mellitus and identify preventive therapeutic targets for treating diabetic vasculopathy. These miRs regulate the expression of target genes, significantly affecting VEGF signaling pathways (including VEGFA, MAPK1, MAPK14, and PIK3CA), insulin signaling, and metabolic processes (such as IRS1, IRS2, IRS4, PIK3CA, and MTOR). This regulation contributes to the pathogenesis and vascular complications of diabetes by modulating critical pathways involved in cell growth, proliferation, and survival.
The interaction network generated by MiRNet 2.0 reveals the complexity of these molecular interactions, showing how miRs and their target genes interact within these pathways. By mapping these interactions, the network highlights the central role of specific miRs in orchestrating the molecular events that lead to diabetic vasculopathy. Understanding this intricate web of interactions can aid in identifying key points for therapeutic intervention, making these miRs promising targets for the prevention and treatment of diabetic vasculopathy. Figure 4 visually represents these interactions, underscoring the potential impact of targeting these miRs in clinical applications. This change was made on page 11 of line 372-390.

Round 2
Reviewer 2 Report
Comments and Suggestions for Authors
I have no further comments.